# Trace metals and their human health risks in sesame seeds from the main cultivation areas of Ethiopia

**Bewketu Mehari** ***, **Tarekegn Fentie Yimer, Tihitna Beletkachew, Eyob Alem, Worku Negash, Mengistu Mulu, Dereje Yenealem, Ayalnesh Miretie**

College of Natural and Computational Sciences, University of Gondar, Gondar, Ethiopia

* bewketu.mehari@uog.edu.et

## Abstract

Sesame (*Sesamum indicum* L.) is a major oilseed crop globally, and white sesame is a key contributor to the foreign exchange earnings of Ethiopia. The main production districts of white sesame in Ethiopia are Humera, Metema, Tegedie, Mirab-Armachiho and Tachi-Armaciho. This study assessed the levels of trace metals (Fe, Cu, Zn, Mn and Ni) in white sesame seeds from these regions and evaluated the associated health risks to consumers. A total of 53 samples were collected from 19 farmer villages across the five districts. Homogenized samples from each village were analyzed using the acid digestion method followed by flame atomic absorption spectroscopy (FAAS). The limit of detection of the method ranged from 0.75 to 865 mg/kg, and the limit of quantitation ranged from 2.55 to 28.8 mg/kg for the different elements analyzed. The recovery of the method was in the range of 90.9–99.6%. The results showed trace metal levels ranging from $164 \pm 6$ to $381 \pm 4$ mg/kg for Fe, $94.0 \pm 1.9$ to $126 \pm 0.8$ mg/kg for Zn, $11.8 \pm 0.4$ to $14.2 \pm 0.4$ mg/kg for Cu, $11.9 \pm 0.9$ to $15.0 \pm 0.7$ mg/kg for Mn and $16.2 \pm 1.1$ to $21.0 \pm 1.2$ mg/kg for Ni across the production districts. One-way ANOVA revealed significant differences ($p < 0.05$) in trace metal concentrations among the districts, indicating a geographical effect on the trace metal content of sesame. Importantly, the study found no non-carcinogenic health risks from the analyzed metals for either adult or child consumers. These findings suggest that the trace metal levels in the sesame seeds are within safe limits for human consumption.

## Introduction

Sesame (*Sesamum indicum* L.) is an important oil crop [1], with its seeds containing approximately 50% oil by weight [2]. It is the second-largest export product in Ethiopia, following coffee [3]. The country's sesame is primarily grown in the northern regions, particularly in semi-arid agro-ecological zones at altitudes ranging from 500 to 1300 meters above sea level [2]. The Humera and Metema districts are the main producers, known for their high-quality sesame. In the 2019/20 crop season, Ethiopia produced over 2.6 million quintals of sesame, with more than 60% of this production coming from these two districts [4]. Given its economic significance, ensuring the quality and safety of sesame is essential, especially considering the potential health risks posed by trace metals.

**Data availability statement:** All relevant data are within the paper and its Supporting Information files.

**Funding:** The author(s) received no specific funding for this work.;

**Competing interests:** The authors have declared that no competing interests exist.

The chemical composition of plants, including sesame, is influenced by various factors such as genetic makeup, environmental conditions, and agricultural practices [5]. Previous studies have shown that the elemental content of crops like coffee [6,7], teff [8] and rice [9] varies significantly between different production regions in Ethiopia, highlighting the geographical influence on trace metal content. This geographical variation is important because it can impact the nutritional and functional properties of sesame, which are vital for both consumers and producers.

Trace metals, while naturally occurring in the environment, can be elevated due to industrialization and agricultural practices [10]. The use of fertilizers, metal-based pesticides, and industrial emissions may contribute to trace metal contamination in crops [11]. These metals, some of which are essential for human health (e.g., Fe, Zn, Cu, Mn), can become toxic when consumed in excessive amounts [12]. Therefore, ensuring that sesame is free from harmful levels of trace metals is crucial for food safety and quality assurance [13,14].

Although previous research on Ethiopian sesame has primarily focused on comparing the physicochemical properties of different sesame varieties [15,16] or analyzing the chemical composition of sesame from a single production district [17], there has been no comprehensive study on the trace metal content of white sesame from all the major production areas. This gap in the literature underscores the need for a broader investigation.

Therefore, the aim of this study is to examine the geographical variations in trace metal content in white sesame seeds produced across Ethiopia's key commercial production areas and assess the associated health risks.

## Materials and methods

### Sample collection

Sesame seed samples were collected from the primary sesame-producing districts of Ethiopia, as illustrated in Fig 1. A total of 53 samples were gathered from 19 farmer villages, known locally as "kebeles," across five production districts. The sampled districts included Metema (Delelo, Meka, Lencha, Metema 01, and Kokit kebeles), Humera (Mikadira and Bereket kebeles), Wolkait (Dansha, Anbaba, Soroka Akafay, Tegedie Ergoye, Tegedie Harid, Tegedie Anbagenet, Tegedie Habtom, and Tegedie Misgan kebeles), Mirab-Armachiho (Abrehajera, Abderafie, and Terefwork kebeles), and Tach-Armachiho (Sanja and Asherie kebeles). Two to three 250 g samples of white sesame seeds were collected from each village, randomly from different farmers. Samples were collected from private farmlands, after permission was granted by the farmers to access the fields. The individual samples from each village were then combined and homogenized before analysis. All samples were collected during the 2023 harvest season.

### Description of the study area

The study area was analyzed using Landsat satellite imagery from the USA Geological Survey (https://earthexplorer.usgs.gov/) with a spatial resolution of 30 meters to generate a land use and land cover thematic map, revealing that approximately 20.2% of the land is used for crop production (Table 1 & Fig 2). A rainfall map was created by interpolating 20 years of average annual rainfall data from nine stations (Table 2) using Spline interpolation in ArcGIS, which indicated annual rainfall value ranging from 120 to 374 mm across the sampling region (Fig 3). The soils in the area are primarily derived from crystalline, volcanic, and Mesozoic sedimentary rocks, as illustrated in Fig 4.

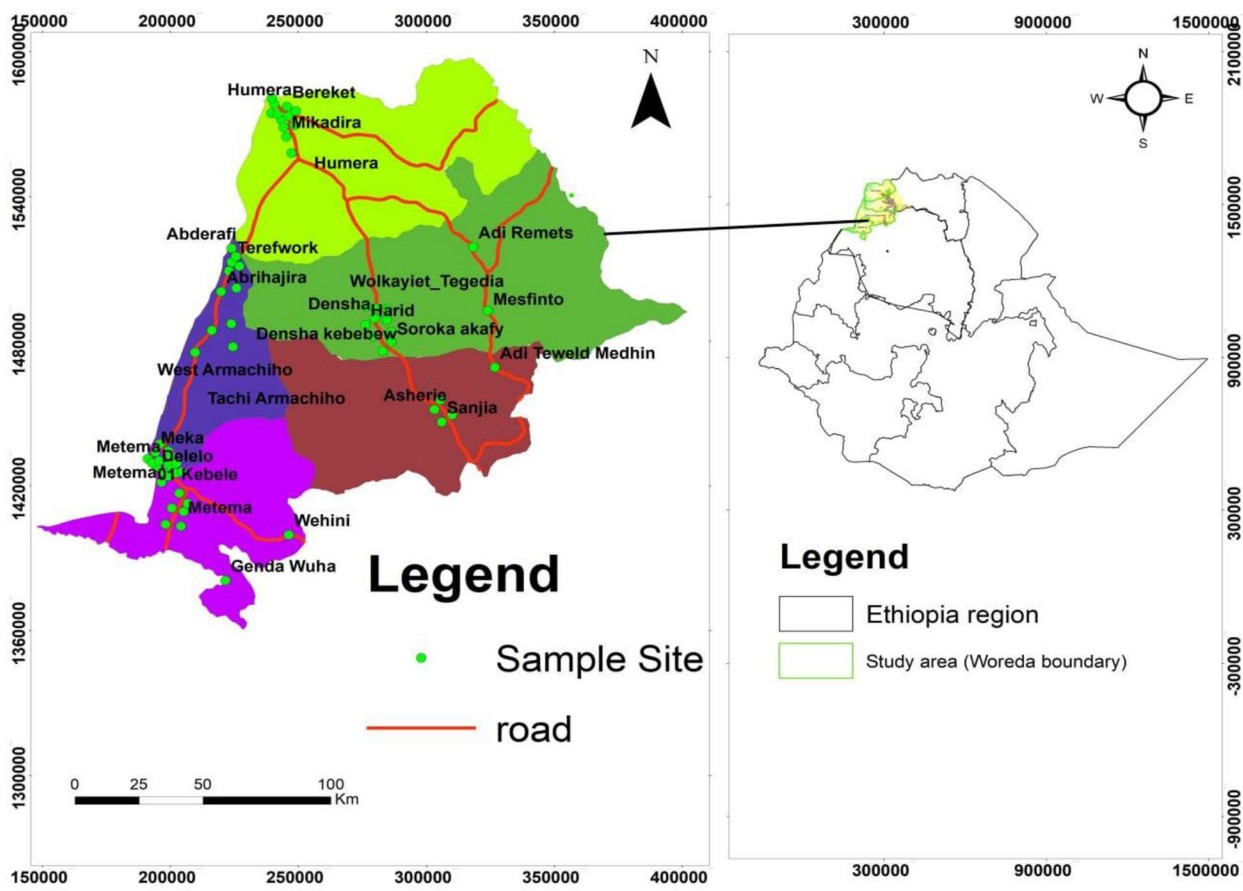

**Fig 1. Map of Ethiopia showing the sesame seed sampling areas.** The map is constructed from a freely available world shape file. ([https://data-catalog.worldbank.org/search/dataset/0038272/World-Bank-Official-Boundaries](https://data-catalog.worldbank.org/search/dataset/0038272/World-Bank-Official-Boundaries)).

**Table 1. Land use coverage in the study area.**

| Land use | Crop | Grass | Forest | Shrub | Settlement |
|---|---|---|---|---|---|
| %Proportion | 20.2% | 16.0% | 38.3% | 25.5% | 0.05% |

## Chemicals and instruments

Nitric acid (69%), $HClO_4$ (70%), $H_2SO_4$ (98%) and $K_2Cr_2O_7$, standard solutions (1000 mg/L) of Fe, Cu, Zn, Mn, and Ni (Blulux Fine Chem, India), and Flame Atomic Absorption Spectrometer (FAAS) (Buck Scientific Model 210VGP, USA) were used in the study.

## Digestion of sesame seeds

Sesame seed samples were first finely powdered and sieved. Then, wet digestion was performed by using a mixture of concentrated $HNO_3$ and $HClO_4$ acids (3:1 v/v). The digestion procedure was optimized following a reported method [17], where a 0.5 g of powdered sesame was mixed with 3 mL of concentrated $HNO_3$ and 2 mL of $HClO_4$ and heated at 200 °C on a hot plate for 50 min to get a clear solution. Then it was filtered through Whatman No 1 filter paper in to a 25 mL volumetric flask and made up to the mark by deionized water.

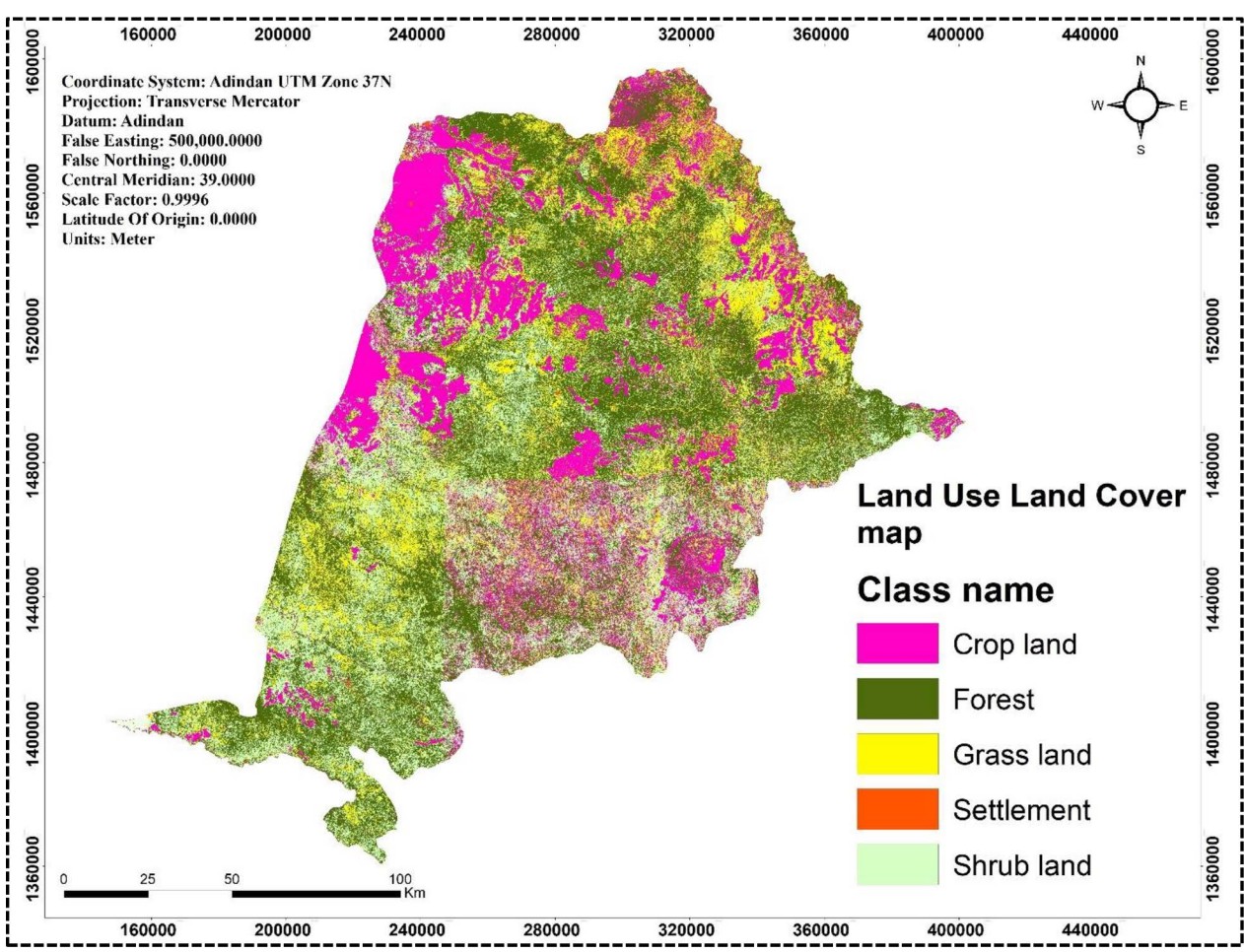

**Fig 2. Land use land cover map of the study area.** The map is constructed from freely available Landsat satellite imagery from the USA Geological Survey (https://earthexplorer.usgs.gov/).

**Table 2. Metrological stations in the study area.**

| Station Name | District | Eelevation (m) | Easting | Northing |
|---|---|---|---|---|
| Humera | Humera | 592 | 240697 | 1578402 |
| Adi Remets | Wolkait | 1983 | 318360 | 1519161 |
| Abderafi | Mirab-Armachiho | 645 | 223933 | 1518553 |
| Mesfinto | Tegedie | 1458 | 323999 | 1492779 |
| Dansha | Tegedie | 888 | 279770 | 1489560 |
| Adi Teweld Medhin | Tach-Armachiho | 2380 | 326793 | 1469111 |
| Metema | Metema | 820 | 191246 | 1431293 |
| Wehini | Metema | 1036 | 246291 | 1399713 |
| Genda Wuha | Metema | 704 | 221528 | 1380770 |

## FAAS Determination of metals

After adjusting the FAAS instrument operating conditions (Table 3), a series of standard solution and the digested sesame sample solutions were submitted for readings of absorbance. Blank samples, which contained only reagents and were digested in the same manner as the

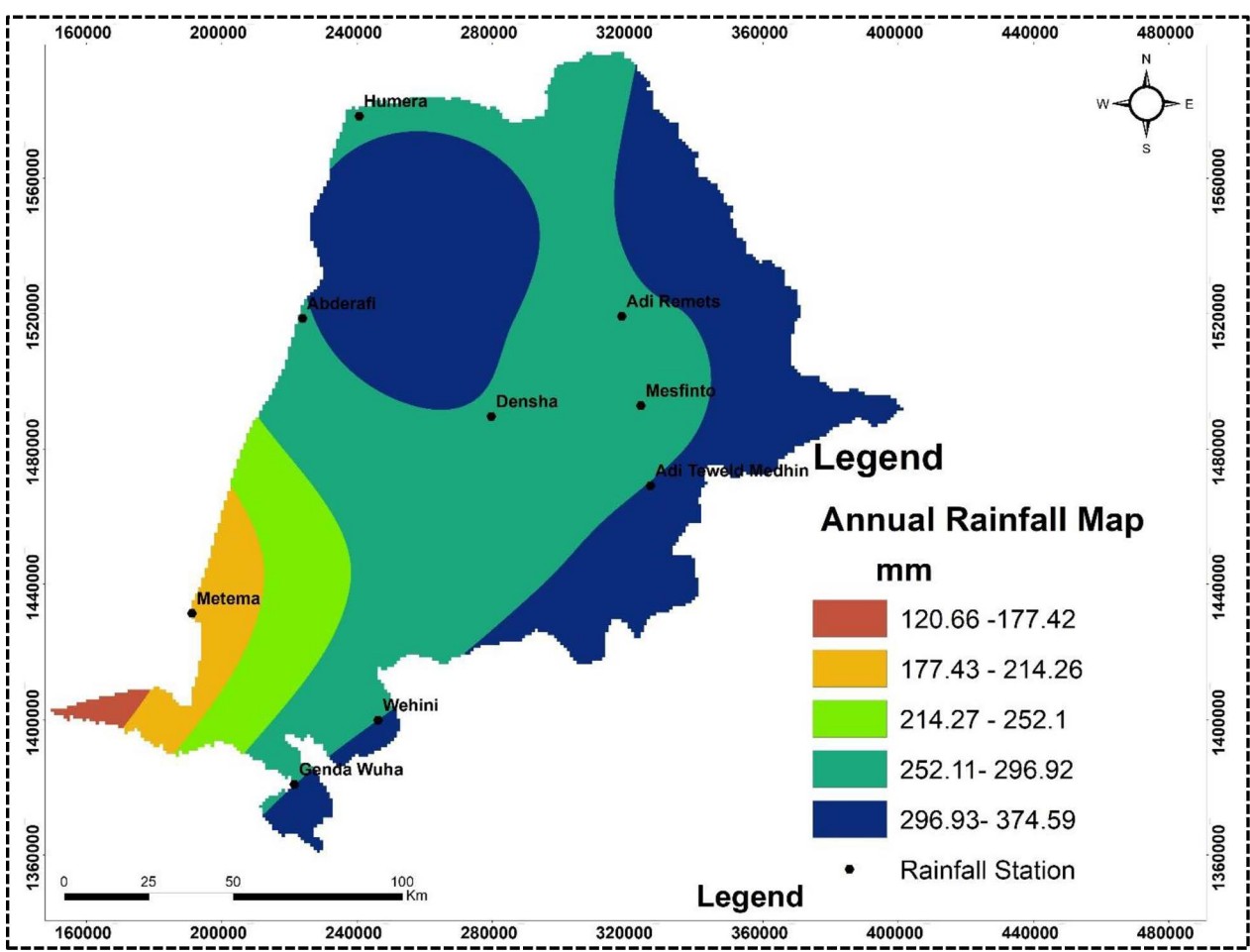

**Fig 3. Rainfall map of the study area.** The map is constructed from a freely available world shape file. (https://datacatalog.worldbank.org/search/dataset/0038272/World-Bank-Official-Boundaries).

samples, were also analysed alongside the samples to account for the baseline response of the instrument. The concentrations of the standard solutions, correlation coefficients ($R^2$), and calibration equations used for the quantification of the metals are provide in Table 4.

### Method validation

The performance of the method was validated using linearity, precision, accuracy, limit of detection (LOD), and limit of quantification (LOQ). The linearity of the method was examined by the coefficient of determination ($R^2$) of each metal obtained from the calibration curve. The accuracy of the method was assessed from the recovery analysis of spiked samples [18]. The relative standard deviation (RSD) values was used to assess the precision of the method with triplicate analysis. The LOD values were calculated as three times, while LOQ ten times, the standard deviation of blank signals divided by the slope of the calibration equation.

### Health risk assessment

The human health risk of consumption of sesame seeds contaminated with heavy metals (Fe, Cu, Mn, Zn, and Ni) was assessed based on estimated daily intake (EDI), target hazard quotient (THQ) and hazard index (HI). The EDI values of metals were calculated to estimate the

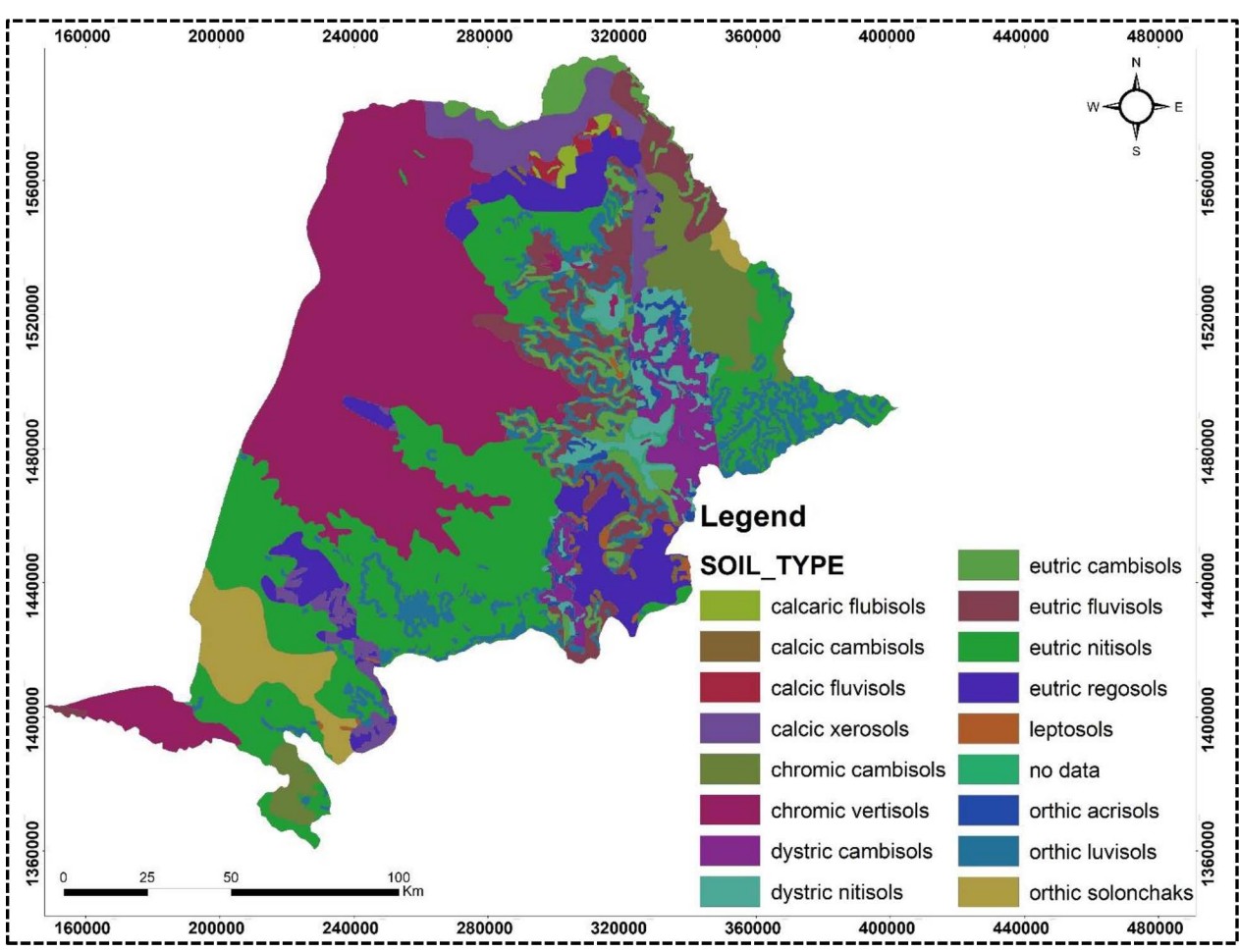

**Fig 4. Soil map of the study area.** The map is constructed from a freely available world shape file. ([https://datacatalog.worldbank.org/search/dataset/0038272/World-Bank-Official-Boundaries](https://datacatalog.worldbank.org/search/dataset/0038272/World-Bank-Official-Boundaries)).

**Table 3. Instrumental operating conditions for the FAAS determinations of metals using air-acetylene flame.**

| Metal | λ (nm) | SW (nm) | I (mA) | Energy (erge) | PMT(v) |
|-------|--------|---------|--------|---------------|--------|
| Fe | 248.3 | 0.2 | 7.0 | 3.108 | 323.5 |
| Cu | 324.7 | 0.7 | 1.5 | 3.844 | 272.0 |
| Mn | 279.5 | 0.7 | 3.0 | 4.11 | 257.0 |
| Zn | 213.9 | 0.7 | 2.0 | 3.201 | 272.9 |
| Ni | 232.01 | 0.2 | 3.0 | 3.103 | 367.4 |

*PMT is photomultiplier tube, SW is slit width, λ is wavelength, I is lamp current.

daily exposure to metals in the body system for a given body weight of a consumer calculated as [19].

$$EDI = \frac{C_{Metal} \times IR}{BW}$$

where $C_{Metal}$ is the mean concentration (mg kg$^{-1}$) of the analyzed metals in the sesame seeds, IR (ingestion rate) is the average daily consumption of sesame (g/day person), and BW is the

**Table 4. The concentrations of standards solutions, calibration equations and correlation coefficients ($R^2$) of metals.**

| Metals | Concentration (mg/L) | $R^2$ | Calibration equation |
|---|---|---|---|
| Fe | 0.05, 0.5, 1, 2.5, 5, 7.5, 10, 15, 20 | 0.9997 | A = 0.01C − 0.0012 |
| Cu | 0.05, 0.5, 1, 2.5, 5, 7.5, 10 | 0.9999 | A = 0.1067C − 0.003 |
| Mn | 0.05, 0.1, 0.5, 1, 2.5, 5, 7.5, 10 | 0.9998 | A = 0.0544C − 0.0025 |
| Zn | 0.05, 0.5, 1, 2.5, 5, 7.5, 10 | 0.9999 | A = 0.1087C + 0.0089 |
| Ni | 0.01, 0.05, 0.5, 1, 2.5, 5, 7.5, 10, 15 | 0.9994 | A = 0.0062C − 0.0007 |

average body weight (kg). The average body weights of adults in Ethiopia was considered 60 kg and its daily intakes of sesame seed for adult was considered 15 g/day person [20]. For children, the average body weight was considered 20 kg and daily consumption was considered 3 g/day person [21].

THQ was applied to assess the non-carcinogenic risk of the metals, which has been proposed by the United States Environmental Protection Agency (USEPA). It refers to the quotient of estimated daily intake (EDI) to the reference dose [22].

$$THQ = \frac{EDI}{RfD}$$

The reference dose value (RfD) of Cu, Zn, Fe, Ni, and Mn are 0.04, 0.30, 0.70, 0.02, and 0.14 mg kg$^{-1}$/day, respectively [19,21–24]. A THQ value greater than 1 indicates that the population poses a non-carcinogenic health risk associated with the consumption of metal contaminated food crops.

Hazard index (HI) was used to estimate the overall non-cancer risk associated with the consumption of multiple potentially toxic heavy metals in the sesame seeds. It is the sum of the hazard quotient (THQ) values of all the metals [20].

$$HI = \sum_{n=i}^{i} THQ$$

When THQ > 1 and HI > 1, there is a possibility of non-carcinogenic health effects [23,24].

### Data Analysis

Data analysis was conducted using STATA software version 14 (StataCorp, USA). The mean and standard deviation were calculated to describe the metal concentrations in the sesame seeds. One-way ANOVA was applied to assess significant differences between samples from different cultivation areas. Before the application of One-way ANOVA, the assumption on normality of the data was checked using the Shapiro-Wilk test and the homogeneity of variances using Levene's test. Pearson correlation coefficients and principal component loadings were used to evaluate the correlation among the trace metals. Statistical significance was set at $p < 0.05$.

## Results and discussion

### Method performance

The accuracy and precision of the method were good enough for the analysis of the elements, with recovery values in the range of 90.9–99.6% and RSD 0.2–10.5% (Table 5). Additionally, the method exhibited good linearity, with coefficient of determination ($R^2$) ≥ 0.9994. The LOD and LOQ were in the range of 0.75–8.65 mg kg$^{-1}$ and 2.55–28.8 mg kg$^{-1}$, respectively.

## Concentration of heavy metals in sesame seeds

**Iron.** The concentration of Fe ranged from $47.7 \pm 2.9$ to $741 \pm 5.0$ mg kg$^{-1}$, with the highest and lowest concentrations recorded for sesame from Metema (Mesheha Lencha) and Tegedie (Dansha Anbaba) sampling regions, respectively (Table 6). Iron is vital for humans as it is involved in oxygen transport and metabolism [13]. In previous studies, the amount of iron in sesame grown in some areas of Ethiopia were reported in the range 12.6-12.9 mg kg$^{-1}$ [18], 36.0-41.6 mg kg$^{-1}$ [13], 30.6–45.9 mg kg$^{-1}$ [20], respectively. While, in different countries the concentration of Fe in sesame seed was found to be 0.0288-0.844 mg kg$^{-1}$ in Iran [21], 65-70 mg kg$^{-1}$ in Brazil [25], and 104-106 mg kg$^{-1}$ in Saudi Arabia [26] (Table 6). The levels of Fe found in this study are higher compared to the previously reported values.

**Copper.** The concentration of Cu ranged from $6.40 \pm 0.2$ to $15.9 \pm 0.5$ mg kg$^{-1}$. The highest and lowest concentrations of Cu were observed in seeds from Metema (Meka) and Tegedie (Dansha Anbaba) sampling areas, respectively (Table 6). When compared with previously reported data in different parts of Ethiopia, Cu was recorded 15.3-18.9 mg kg$^{-1}$ [13], 11.28–17.60 mg kg$^{-1}$ [20]. The results of this study are, therefore, comparable with the literature

**Table 5. The recovery (%), relative standard deviation (%), coefficient of determination ($R^2$), limit of detection (LOD) and limit of quantitation (LOQ) of the FAAS method used to determine trace metals in sesame seeds.**

| Element | Recovery (%) | RSD (%) | $R^2$ | LOD (mg kg$^{-1}$) | LOQ (mg kg$^{-1}$) |
|---|---|---|---|---|---|
| Fe | 90.9 | 0.5-8.1 | 0.9997 | 8.65 | 28.8 |
| Mn | 99.6 | 0.6-9.4 | 0.9998 | 1.45 | 4.90 |
| Ni | 95.6 | 1.7-10.5 | 0.9994 | 2.25 | 6.50 |
| Zn | 90.9 | 0.2-2.3 | 0.9999 | 7.35 | 24.4 |
| Cu | 95.8 | 0.3-3.9 | 0.9999 | 0.75 | 2.55 |

**Table 6. The concentration (mg kg$^{-1}$) of heavy metals (mean $\pm$ SD) in sesame seed samples.**

| District | Sampling Areas | Fe | Mn | Ni | Zn | Cu |
|---|---|---|---|---|---|---|
| Metema | Metema 01-1 | $62.7 \pm 2.9$ | $8.12 \pm 0.5$ | $22.4 \pm 1.0$ | $107 \pm 0.5$ | $12.8 \pm 0.2$ |
| | Metema 01-2 | $741 \pm 5.0$ | $15.5 \pm 1.4$ | $11.9 \pm 0.4$ | $104 \pm 0.7$ | $14.4 \pm 0.3$ |
| | Delelo | $94.3 \pm 7.6$ | $9.96 \pm 0.5$ | $16.7 \pm 1.7$ | $90.7 \pm 0.5$ | $13.1 \pm 0.5$ |
| | Kokit | $144 \pm 2.9$ | $11.5 \pm 0.9$ | $20.5 \pm 1.2$ | $108 \pm 2.3$ | $15.0 \pm 0.5$ |
| | Meka | $124 \pm 2.9$ | $14.2 \pm 0.9$ | $14.6 \pm 0.8$ | $127 \pm 0.7$ | $15.9 \pm 0.5$ |
| Tach-Armachiho | Sanja | $224 \pm 2.9$ | $13.9 \pm 0.5$ | $22.6 \pm 0.8$ | $92.2 \pm 1.9$ | $14.0 \pm 0.5$ |
| | Asherie | $221 \pm 5.0$ | $13.3 \pm 0.9$ | $19.4 \pm 1.5$ | $95.9 \pm 1.9$ | $12.8 \pm 0.3$ |
| Tegedie | Ergoye | $328 \pm 7.6$ | $20.4 \pm 0.5$ | $8.92 \pm 0.8$ | $164 \pm 1.7$ | $12.3 \pm 0.2$ |
| | Harid | $361 \pm 10$ | $15.5 \pm 1.4$ | $17.8 \pm 0.7$ | $103 \pm 0.8$ | $14.8 \pm 0.3$ |
| | Soroka-1 | $303 \pm 5.8$ | $13.3 \pm 0.9$ | $23.4 \pm 1.7$ | $97.3 \pm 0.7$ | $13.0 \pm 0.3$ |
| | Soroka-2 | $436 \pm 8.3$ | $14.9 \pm 0.5$ | $20.2 \pm 1.5$ | $91.6 \pm 1.4$ | $14.4 \pm 0.3$ |
| | Dansha-1 | $47.7 \pm 2.9$ | $5.67 \pm 0.5$ | $7.31 \pm 0.7$ | $38.8 \pm 0.3$ | $6.42 \pm 0.2$ |
| | Dansha-2 | $328 \pm 10$ | $20.4 \pm 0.5$ | $20.5 \pm 1.2$ | $166 \pm 0.3$ | $13.9 \pm 0.3$ |
| | Dansha-3 | $313 \pm 15$ | $12.4 \pm 0.9$ | $15.4 \pm 0.8$ | $96.8 \pm 2.2$ | $13.0 \pm 0.3$ |
| Mirab-Armachiho | Abrehajera | $341 \pm 10$ | $14.9 \pm 0.5$ | $14.3 \pm 1.0$ | $87.8 \pm 0.3$ | $13.3 \pm 0.2$ |
| | Abderafie | $76.0 \pm 5.0$ | $12.1 \pm 0.5$ | $23.7 \pm 2.4$ | $90.7 \pm 0.9$ | $12.6 \pm 0.5$ |
| | Terefwork | $74.3 \pm 2.9$ | $12.7 \pm 0.5$ | $15.6 \pm 1.2$ | $114 \pm 0.7$ | $13.1 \pm 0.4$ |
| Humera | Mikadira | $614 \pm 2.9$ | $16.1 \pm 0.9$ | $15.4 \pm 1.6$ | $137 \pm 0.7$ | $12.6 \pm 0.5$ |
| | Bereket | $148 \pm 5.8$ | $13.9 \pm 0.5$ | $25.0 \pm 1.7$ | $114 \pm 1.0$ | $10.9 \pm 0.2$ |

values, however it was higher than the result 4.9‒5.0 mg kg⁻¹ reported by Beshaw et al. [19] (Table 7).

In previous studies in different countries, the level of Cu in sesame was reported as 0.226‒0.495 mg kg⁻¹ in Iran [21], and 3.2-7.2 mg kg⁻¹ in Nigeria [27], which are lower than the amounts found in this study. Whereas, relatively comparable results were reported for sesame seeds from Korea (5.9–18.7 mg kg⁻¹) [28] and Brazil (16.0-19.0 mg kg⁻¹) [25] (Table 7).

**Zinc.** The concentration of Zn ranged from $38.8 \pm 0.3$ to $166 \pm 0.3$ mg kg⁻¹ across the different sampling sites. Both the highest and lowest concentrations were measured in sesame seeds from Tegedie, Dansha-2 (Anbagenet area) and Dansha-1 (Anbaba area), respectively. The concentration of zinc in the sesame seed samples was consistence with the result (95.8–120 mg kg⁻¹) reported by Mengistu *et al.* [20] from Ethiopia. However, other literatures reported the levels of Zn in Ethiopian sesame seeds were 8.3-8.7 mg kg⁻¹ [18], and 57.9-61.9 mg kg⁻¹ [13], which were much lower than the current results. Moreover, the studies conducted in sesame seed samples from Iran, Brazil, Nigeria, Saudi Arabia, and Korea were reported as 1.105-2.228 mg kg⁻¹ [21], 69.0-81.0 mg kg⁻¹ [25], 9.0-28.0 mg kg⁻¹ [27], 36.0-38.0 mg kg⁻¹ [26], and 35.2–71.5 mg kg⁻¹ [28], respectively, which were lower than this study.

**Manganese.** The measured concentration of Mn was in the range of $5.67 \pm 0.5$ to $20.4 \pm 0.5$ mg kg⁻¹. Both the highest and lowest concentrations were found in sesame seeds from Tegedie district, Ergoye and Dansha-1 (Anbaba area), respectively. The concentrations of Mn obtained in the sesame samples were lower than that (63.20–75.48 mg kg⁻¹) reported previously [20]. However, relatively consistent results were reported from Brazil [25], Nigeria [27], and Korea [28] that reported 15.0–21.0, 10.0–24.0, and 12.9–33.0 mg kg⁻¹, respectively. The result reported by Eghbaljoo-Gharehgheshlaghi et al. [21] from Iran (0.16‒0.53) mg kg⁻¹ was lower than the results of this study.

**Nickel.** The concentration of Ni ranged from $7.31 \pm 0.7$ to $25.0 \pm 1.7$ mg kg⁻¹. The highest and lowest concentrations were observed in sesame seeds from Bereket (Humera) and Dansha-1 (Tegedie) sampling sites, respectively. In the literature, the concentration of Nickel in sesame seed grown in different countries was found to be 0.0028-0.0697 mg kg⁻¹ in Iran [21], 2.0-6.0 mg kg⁻¹ in Nigeria [27], and 0.507-3.043 mg kg⁻¹ in Korea [28], which are relatively lower than the concentration of Ni found in this study.

## Comparison among production districts

Statistical analysis using one-way ANOVA revealed the presence of significant difference (p 0.05) among districts in the concentrations of the analyzed metals, which indicated geographical effects on the amount of trace metals in the sesame seeds. The highest concentrations of

**Table 7. Comparison of the concentrations of metals in the sesame seeds with literature reports from different countries.**

| Country | Fe | Cu | Zn | Mn | Ni | Ref. |
|---|---|---|---|---|---|---|
| Ethiopia | 47.7-741 | 6.40-15.9 | 38.8-166 | 5.67-20.4 | 7.31-25.0 | This study |
| Ethiopia | 36.0-41.6 | 15.3-18.9 | 57.9-61.9 | – | – | [13] |
| Ethiopia | 12.6-12.9 | 4.9-5.0 | 8.3-8.7 | – | – | [19] |
| Ethiopia | 30.62–45.92 | 11.28–17.60 | 95.8–120.3 | 63.20–75.48 | – | [20] |
| Iran | 0.0288-0.844 | 0.226-0.495 | 1.105-2.228 | 0.161-0.5298 | 0.0028-0.0697 | [21] |
| Brazil | 65.0-70.0 | 16.0-19.0 | 69.0-81.0 | 15.0-21.0 | – | [25] |
| Saudi Arabia | 104.0-106.0 | – | 36.0-38.0 | – | – | [26] |
| Nigeria | – | 3.200-7.200 | 9.00-28.00 | 10.00-24.00 | 2.00-6.00 | [27] |
| Korea | – | 5.9–18.7 | 35.2–71.5 | 12.9–33.0 | 0.507-3.043 | [28] |

Fe, Zn and Mn were found in sesame seeds from Humera (Table 8). On the other hand, higher levels of Ni was observed in seeds from Tachi-Armachio while Cu from Metema.

The differences in trace metal content observed among sesame seeds from different cultivation areas could be attributed to factors such as genetic variation of the plant, environmental factors like soil composition, or agronomic practices, including the type and quantity of fertilizers used. The elemental composition of a plant is generally a reflection of the elemental composition of the soil, however, the accumulation of an element within the plant depends on the nature of the element, plant species, and environmental conditions [6,7].

## Correlation analysis

Correlation analyses play a crucial role in understanding the relationships between trace metals [25], as they help to differentiate the potential sources of these metals, whether naturally occurring in the soil or introduced through anthropogenic activities, such as the application of fertilizers [29]. In this study, iron (Fe) exhibited a strong positive and significant correlation with zinc (Zn) and manganese (Mn) (Table 9), suggesting that their sources may be linked, potentially originating from similar geological or soil-related processes [30]. Conversely, copper (Cu demonstrated a strong negative and significant correlation with all other metals, except for nickel (Ni), which indicates that copper may have distinct sources influencing its concentration in the study area, possibly related to agricultural inputs. Interestingly, nickel (Ni) showed no correlation with any of the other trace elements measured, implying that its source may be unrelated to those of Fe, Zn, Mn, or Cu, and could be linked to specific geological formations or other environmental factors [31]. To further investigate the correlation between the metals, principal component analysis (PCA) was conducted. Two principal components (PCs) were extracted, with PC1 explaining 50% and PC2 explaining 25% of the variance in the dataset. The PCA results revealed that Fe, Mn, Zn, and Cu were strongly correlated with PC1 and grouped together, while Ni was strongly correlated with PC2 (Table 10). Therefore, the distinct patterns of correlation suggest that the sources of Fe, Zn, and Mn may differ from those of Cu and Ni, pointing to the complexity of trace metal interactions in the environment.

**Table 8. Comparison of the concentration (mg kg⁻¹) of heavy metals in sesame seeds among districts (mean±SD).**

| District | Fe | Mn | Ni | Zn | Cu |
|---|---|---|---|---|---|
| Metema | 233 ± 4 | 11.9 ± 0.9 | 17.2 ± 1.0 | 107 ± 0.9 | 14.2 ± 0.4 |
| Tachi-Armachio | 223 ± 4 | 13.6 ± 0.7 | 21.0 ± 1.2 | 94.0 ± 1.9 | 13.4 ± 0.4 |
| Tegedie | 302 ± 9 | 14.6 ± 0.8 | 16.2 ± 1.1 | 108 ± 1.0 | 12.5 ± 0.3 |
| Mirab-Armachiho | 164 ± 6 | 13.2 ± 0.5 | 17.9 ± 1.5 | 97.3 ± 0.6 | 13.0 ± 0.3 |
| Humera | 381 ± 4 | 15.0 ± 0.7 | 20.2 ± 1.6 | 126 ± 0.8 | 11.8 ± 0.4 |

**Table 9. Pearson correlation coefficients among trace metals determined in sesame seeds from the major production districts of Ethiopia.**

| | Fe | Mn | Ni | Zn | Cu |
|---|---|---|---|---|---|
| Fe | 1 | | | | |
| Mn | 0.711* | 1 | | | |
| Ni | 0.164 | 0.258 | 1 | | |
| Zn | 0.904* | 0.498 | 0.026 | 1 | |
| Cu | -0.703* | -0.943* | -0.209 | -0.629* | 1 |

*The correlation is significant at p < 0.05.

**Table 10. Loadings obtained from the principal component analysis of the trace metals in the sesame seeds.**

| Element | Component 1 | Component 2 |
|---|---|---|
| Fe | 0.663 | -0.460 |
| Mn | 0.932 | -0.114 |
| Ni | 0.049 | 0.920 |
| Zn | 0.838 | 0.069 |
| Cu | -0.692 | 0.444 |

## Health risk assessment

**Estimated dietary intake.** In the assessment of human health risks associated with the consumption of potentially toxic trace metals in food, the most widely used procedure is the calculation of estimated dietary intake (EDI) values and their comparison with standard recommended dietary intake (RDI) values [14]. The WHO/FAO recommended RDI values of Fe, Cu, and Zn has been 9-17, 2-3, and 20 mg/day for adults, respectively [32]. While, RDI of Mn for adult consumers is 2.5–5.0 mg/day. Similarly, the provisional tolerable daily intake (PTDI) value for Ni is 0.3 mg/day [14].

The calculated EDI values of Fe, Cu, Zn, Mn, and Ni, in all of the sesame seed samples, for adults ranged from 0.0409-0.0953, 0.0029-0.0036, 0.0235-0.0315 0.0030-0.0038, 0.0041-0.0053 mg/day, respectively (Table 11). While, for children the calculated EDI values of Fe, Cu, Zn, Mn, and Ni in sesame seeds were ranged from 0.0246-0.0572, 0.0018-0.0021, 0.0141-0.0189, 0.0018-0.0023, 0.0024-0.0032 mg/day, respectively. Accordingly, the calculated EDI values of all the analyzed trace metals in sesame seeds, for both adults and children, were found to be below the WHO/FAO recommended daily intake values.

## Hazard quotient and hazard index

Hazard quotient (HQ) is a measure of non-carcinogenic health risks associated with a trace metal, while hazard index (HI) is used to estimate the overall non-cancer risk associated with the consumption of multiple potentially toxic heavy metals [33]. The standard guideline values of both HQ and HI have been reported to be ≤1.0 [33]. In this regard, both the HQ values, for the individual elements, and HI, for the combined elements, were below the standard guideline value of unity (Table 12), which suggested the absence of associated non-carcinogenic human health risks with the consumption of the studied white sesame seeds.

This study was conducted considering of trace metals that could cause a serious health risk when their levels exceed the maximum permissible limits [20]. The essential metals may create toxic effects when taken in high amounts [14]. However, in this study all the analyzed heavy metals revealed no human health risks from the trace metals present in the sesame seeds.

**Table 11. Estimated daily intake (EDI) values (mg/person/day) of heavy metals in sesame seeds from the major production areas of Ethiopia.**

| Districts | EDI (mg/person/day) | | | | | | | | | |
|---|---|---|---|---|---|---|---|---|---|---|
| | Adult | | | | | Child | | | | |
| | Fe | Mn | Ni | Zn | Cu | Fe | Mn | Ni | Zn | Cu |
| Metema | 0.058 | 0.003 | 0.004 | 0.027 | 0.004 | 0.035 | 0.002 | 0.003 | 0.016 | 0.002 |
| Tach-Armachio | 0.056 | 0.003 | 0.005 | 0.024 | 0.003 | 0.033 | 0.002 | 0.003 | 0.014 | 0.002 |
| Tegedie | 0.076 | 0.004 | 0.004 | 0.027 | 0.003 | 0.045 | 0.002 | 0.002 | 0.016 | 0.002 |
| Mirab-Armachiho | 0.041 | 0.003 | 0.005 | 0.024 | 0.003 | 0.025 | 0.002 | 0.003 | 0.015 | 0.002 |
| Humera | 0.095 | 0.004 | 0.005 | 0.032 | 0.003 | 0.057 | 0.002 | 0.003 | 0.019 | 0.002 |

**Table 12. The calculated hazard quotient (HQ) and hazard index (HI) values of trace metals in sesame seeds from the major production areas of Ethiopia.**

| Districts | Adult | | | | | | Child | | | | | |
|---|---|---|---|---|---|---|---|---|---|---|---|---|
| | HQ | | | | | HI | HQ | | | | | HI |
| | Fe | Mn | Ni | Zn | Cu | | Fe | Mn | Ni | Zn | Cu | |
| Metema | 0.08 | 0.02 | 0.22 | 0.09 | 0.09 | 0.50 | 0.05 | 0.01 | 0.13 | 0.05 | 0.05 | 0.30 |
| Tach-Armachio | 0.08 | 0.02 | 0.26 | 0.08 | 0.08 | 0.53 | 0.05 | 0.02 | 0.16 | 0.05 | 0.05 | 0.32 |
| Tegedie | 0.11 | 0.03 | 0.20 | 0.09 | 0.08 | 0.51 | 0.07 | 0.02 | 0.12 | 0.05 | 0.05 | 0.30 |
| Mirab-Armachiho | 0.06 | 0.02 | 0.22 | 0.08 | 0.08 | 0.47 | 0.04 | 0.01 | 0.13 | 0.05 | 0.05 | 0.28 |
| Humera | 0.14 | 0.03 | 0.25 | 0.11 | 0.07 | 0.59 | 0.08 | 0.02 | 0.15 | 0.06 | 0.04 | 0.36 |

# Conclusion

In this study, the concentrations of trace metals (Fe, Cu, Zn, Mn, and Ni) in white sesame seeds were found to vary significantly across different production areas in Ethiopia. Iron, Zn, and Mn exhibited significant positive correlations with one another, indicating that these metals likely share a common source, such as the underlying geology or applied chemical fertilizers. All measured concentrations of trace metals were below the established standard guideline values, and no non-carcinogenic health risks were identified for either adult or child consumers. These findings suggest that the trace metal levels in the sesame seeds are within safe limits for human consumption. The study did not analyze the soil composition or the specific types of fertilizers used in the cultivation areas, which are essential for understanding the sources of trace metals and their potential impact on crop contamination. While the study found no non-carcinogenic health risks based on the measured trace metal concentrations, it focused only on a limited set of metals and did not account for other potentially toxic metals that could also pose health risks. All of these are recommended for future studies.

# Acknowledgments

The authors thank University of Gondar for giving access to its laboratory facility.

# Author contributions

**Conceptualization:** Bewketu Mehari.

**Data curation:** Worku Negash, Mengistu Mulu, Dereje Yenealem, Ayalnesh Miretie.

**Investigation:** Tarekegn Fentie Yimer, Tihitna Beletkachew, Eyob Alem.

**Methodology:** Bewketu Mehari.

**Writing – review & editing:** Bewketu Mehari.

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
