## [Decision Letter · Decision Letter 0]

7 Nov 2024

PONE-D-24-46643Trace metals and their human health risks in sesame seeds from the main cultivation areas of EthiopiaPLOS ONE

Dear Dr. Mehari,

Thank you for submitting your manuscript to PLOS ONE. After careful consideration, we feel that it has merit but does not fully meet PLOS ONE’s publication criteria as it currently stands. Therefore, we invite you to submit a revised version of the manuscript that addresses the points raised during the review process.

We look forward to receiving your revised manuscript.

Kind regards,

Muhammad Sibt-e-Abbas

Academic Editor

PLOS ONE

**Journal Requirements:**

5. We note that Figures 1 to 4 in your submission contain map images which may be copyrighted. All PLOS content is published under the Creative Commons Attribution License (CC BY 4.0), which means that the manuscript, images, and Supporting Information files will be freely available online, and any third party is permitted to access, download, copy, distribute, and use these materials in any way, even commercially, with proper attribution. For these reasons, we cannot publish previously copyrighted maps or satellite images created using proprietary data, such as Google software (Google Maps, Street View, and Earth). For more information, see our copyright guidelines: http://journals.plos.org/plosone/s/licenses-and-copyright.

We require you to either present written permission from the copyright holder to publish these figures specifically under the CC BY 4.0 license, or remove the figures from your submission:

a. You may seek permission from the original copyright holder of Figures 1 to 4 to publish the content specifically under the CC BY 4.0 license.  

Reviewers' comments:

Reviewer's Responses to Questions

**Comments to the Author**

1. Is the manuscript technically sound, and do the data support the conclusions?

Reviewer #1: Yes

Reviewer #2: No

Reviewer #3: Partly

2. Has the statistical analysis been performed appropriately and rigorously? 

Reviewer #1: Yes

Reviewer #2: Yes

Reviewer #3: No

3. Have the authors made all data underlying the findings in their manuscript fully available?

Reviewer #1: Yes

Reviewer #2: Yes

Reviewer #3: No

4. Is the manuscript presented in an intelligible fashion and written in standard English?

Reviewer #1: Yes

Reviewer #2: Yes

Reviewer #3: No

5. Review Comments to the Author

**Reviewer #1:**  Comments to the Author √

The authors performed "Trace metals and their human health risks in sesame seeds from the main cultivation areas of Ethiopia" in Journal of PLOS ONE. The methodology is good. I suggest major revisions for the article before its publication the Journal of PLOS ONE. In my opinion, the English language expressions in the manuscript require copyediting. The writing of this manuscript is not easy for readers to understand. Some sentences fail to express the original meaning. I suggest using a professional copyeditor or a native English speaker to copy edit this manuscript.

Abstract:

Abstract section should be clear and concise.

Where is LOD, LOQ, recovery?

Key word: Sort by alphabetical order.

Introduction

The logic of the current introduction should be revised, and I suggest organizing the Introduction section as following order: importance and meanings, previous studies (literature review), the gaps of previous studies, and objectives of this study.

The text has many errors that must be carefully corrected by the author

The novelty of the research is not justified.

Introduction is not including up to date and relevant articles

Please see the following articles and to make the article more complete, they must be added to the text

Assessment of heavy metal content in refined and unrefined salts obtained from Urmia, Iran.

Determination of Element Concentration of Brewed Tea Consumed in Iran Using ICP-OES: A Risk Assessment Study

Assessment of rice marketed in Iran with emphasis on toxic and essential elements; effect of different cooking methods

The concentration and probabilistic health risk assessment of trace metals in three type of sesame seeds using ICP-OES in Iran

Methodology

Material and method section is not providing sufficient explanation

Complete and accurate data analysis should be mentioned

Statistical analysis should be clear and concise.

Result and Discussion

Please mention the validation method in the text of the article

For the results to be written more carefully, please pay attention to the mentioned content and articles

It is necessary to compare with more studies and summarize the results of other studies.

Recommendations and Future Implications should be clear and concise

Conclusion

Conclusions should be clear and concise

Please write the limitations of the study

Throughout the text, some words must match the format of the journal.

In general, the text has many errors that must be carefully corrected by the author.

Use newer references

**Reviewer #2:**  In this article, the concentration of some heavy metals in sesame seeds harvested from different farms in Ethiopia has been investigated. Although the topic of the article is a priority in terms of the importance of food health, but the article has flaws that are not acceptable in my opinion. These are listed below.

1- The introduction does not sufficiently specify the importance and innovation of this article.

2- Chemical fertilizers used in different regions are not mentioned in the materials and method section.

3- It seems that checking and analyzing the concentration of heavy elements alone in sesame seeds is not enough. It is suggested that the concentration of heavy metals in the soil of the region was also checked.

4- The results part seems to be more like presenting a scientific report rather than a scientific and research article.

5- What is the purpose of statistical correlation analysis? What is the conclusion of the presentation of correlation results?

6- In Table 9, the correlation analysis results of significant levels are not specified.

7- It is suggested to use more statistical analyzes such as principal component analysis in addition to correlation to determine the concentration of heavy metals.

8- The conclusion should be fundamentally rewritten. The results of this paper should be compared and evaluated with other similar researches in Ethiopia or other regions or other products.

9- What is the reason for the presence of heavy metals in sesame seeds? This is important to discuss.

**Reviewer #3:**  The manuscript deals with "Trace metals and their human health risks in sesame seeds from the main cultivation areas of Ethiopia" the following comments and suggestions are need to address before its consideration.

1. The abstract should succinctly summarize the key findings, including specific health risks associated with trace metals. Currently, it mentions that non-carcinogenic health risks were not observed, but elaborating on the implications of these findings would strengthen the abstract

2. The methodology section should provide more detail on the sampling process, including the number of samples collected from each district and the criteria for selection. This will enhance reproducibility and transparency in the research

3. The introduction highlights the geographical influence on trace metal contents, but the discussion could further explore how specific environmental factors in each district may contribute to the observed differences in metal concentrations. This would provide a more comprehensive understanding of the results

4. While the statistical methods used (e.g., one-way ANOVA) are mentioned, a more thorough explanation of how these analyses were conducted and interpreted would be beneficial. Including assumptions of the tests and how they were checked could improve the rigor of the analysis

5. More emphasis on the quality control measures taken during the study, such as the use of blank solutions and recovery studies, would add credibility to the results. This could be integrated into the methodology section. https://link.springer.com/article/10.1007/s12011-024-04234-0,

https://www.sciencedirect.com/science/article/pii/S0278691524003727,

6. 1. What specific AAS techniques were utilized?, 2. How do findings compare to previous studies? 3. What quality control measures were implemented? Referred to below papers and cite them

https://www.sciencedirect.com/science/article/pii/S0946672X24001019, https://www.sciencedirect.com/science/article/pii/S0946672X24000749, https://www.sciencedirect.com/science/article/pii/S0889157523008177

7. The manuscript includes comparisons with previous studies, but it could benefit from a more systematic comparison of trace metal levels across different countries. This would contextualize the findings within a global framework and highlight the significance of the results

8. The health risk assessment section should include a more detailed discussion on the potential long-term effects of trace metal consumption, particularly for vulnerable populations such as children. This would enhance the relevance of the study to public health

9. Incorporating graphs or tables to visually represent the concentration levels of trace metals across different regions would improve the readability and impact of the findings. Visual aids can help convey complex data more effectively

10. The conclusion should not only summarize the findings but also suggest practical implications for consumers and policymakers regarding sesame seed consumption and agricultural practices. This would provide actionable insights based on the research

6. PLOS authors have the option to publish the peer review history of their article (what does this mean? ). If published, this will include your full peer review and any attached files.

**Do you want your identity to be public for this peer review?** For information about this choice, including consent withdrawal, please see our Privacy Policy .

Reviewer #1: No

Reviewer #2: No

Reviewer #3: No

---

## [Author Response · Author response to Decision Letter 1]

22 Dec 2024

PONE-D-24-46643

Trace metals and their human health risks in sesame seeds from the main cultivation areas of Ethiopia

Dear Muhammad Sibt-e-Abbas

Academic Editor

PLOS ONE,

Author Response: We thank the editor and the three reviewers for reviewing our manuscript favorably.

Journal Requirements:

Author Response: We thank the editor for handling our manuscript favorably. We have followed PLOS ONE’s style requirements.

Author Response: Samples were collected from private farmlands, after permission was granted by the farmers to access their fields. This information is now provided in the method section.

Author Response: We confirm that the submission contains all raw data required to replicate the results of the study.

Author Response: We are happy to comply with the open data policy of PLOS ONE. Could you kindly suggest the statement that best clarifies this?

5. We note that Figures 1 to 4 in your submission contain map images which may be copyrighted. All PLOS content is published under the Creative Commons Attribution License (CC BY 4.0), which means that the manuscript, images, and Supporting Information files will be freely available online, and any third party is permitted to access, download, copy, distribute, and use these materials in any way, even commercially, with proper attribution. For these reasons, we cannot publish previously copyrighted maps or satellite images created using proprietary data, such as Google software (Google Maps, Street View, and Earth). For more information, see our copyright guidelines: http://journals.plos.org/plosone/s/licenses-and-copyright.

Author Response: The maps in Figure 1, 2 and 4 are constructed from a freely available world shape file (https://datacatalog.worldbank.org/search/dataset/0038272/World-Bank-Official-Boundaries). The Maps in Figure 3 is constructed from freely available Landsat satellite imagery from the USA Geological Survey (https://earthexplorer.usgs.gov/). These are now indicated in the figure captions.

We require you to either present written permission from the copyright holder to publish these figures specifically under the CC BY 4.0 license, or remove the figures from your submission:

a. You may seek permission from the original copyright holder of Figures 1 to 4 to publish the content specifically under the CC BY 4.0 license.

Reviewers' comments:

Reviewer #1

The authors performed "Trace metals and their human health risks in sesame seeds from the main cultivation areas of Ethiopia" in Journal of PLOS ONE. The methodology is good. I suggest major revisions for the article before its publication the Journal of PLOS ONE. In my opinion, the English language expressions in the manuscript require copyediting. The writing of this manuscript is not easy for readers to understand. Some sentences fail to express the original meaning. I suggest using a professional copyeditor or a native English speaker to copy edit this manuscript.

Author Response: We thank Reviewer #1 for the positive review of our manuscript. In response, we have thoroughly revised the manuscript to address language issues.

Abstract:

Abstract section should be clear and concise.

Where is LOD, LOQ, recovery?

Author Response: We have revised the abstract and included LOD, LOQ and recovery values.

Key word: Sort by alphabetical order.

Author Response: We have sorted the keywords by alphabetical order.

Introduction

The logic of the current introduction should be revised, and I suggest organizing the Introduction section as following order: importance and meanings, previous studies (literature review), the gaps of previous studies, and objectives of this study.

Author Response: We have revised the introduction based on the suggestion.

The text has many errors that must be carefully corrected by the author

Author Response: We have corrected the introduction by addressing any typographical and language errors.

The novelty of the research is not justified.

Author Response: The novelty of this research lies in providing insights into the trace metal content of white sesame seeds from all major sesame-producing regions in Ethiopia, addressing a significant gap in the literature. Unlike previous studies that focused on limited regions, this study examined geographical variations in trace metal levels and evaluated the associated health risks, which are crucial for food safety. This is clearly indicated in the introduction.

Introduction is not including up to date and relevant articles

Please see the following articles and to make the article more complete, they must be added to the text

Assessment of heavy metal content in refined and unrefined salts obtained from Urmia, Iran.

Determination of Element Concentration of Brewed Tea Consumed in Iran Using ICP-OES: A Risk Assessment Study

Assessment of rice marketed in Iran with emphasis on toxic and essential elements; effect of different cooking methods

The concentration and probabilistic health risk assessment of trace metals in three type of sesame seeds using ICP-OES in Iran

Author Response: The article titled “The concentration and probabilistic health risk assessment of trace metals in three type of sesame seeds using ICP-OES in Iran” has already been cited in the manuscript, now as reference [21]. We have also added the article “Assessment of rice marketed in Iran with emphasis on toxic and essential elements; effect of different cooking methods” as reference [10].

Methodology

Material and method section is not providing sufficient explanation

Author Response: We have revised the Material and Method section.

Complete and accurate data analysis should be mentioned

Author Response: We have revised the data analysis section for clarity.

Statistical analysis should be clear and concise.

Author Response: We have revised the data analysis section for clarity.

Result and Discussion

Please mention the validation method in the text of the article

Author Response: The method validation had been provided in the “Material and method” section, under the subsection “Method validation”.

For the results to be written more carefully, please pay attention to the mentioned content and articles

It is necessary to compare with more studies and summarize the results of other studies.

Recommendations and Future Implications should be clear and concise

Author Response: We have compared our results with those reported in the literature to contextualize our findings.

Conclusion

Conclusions should be clear and concise

Author Response: We have revised the conclusion for clarity.

Please write the limitations of the study

Author Response: We have provide the limitations of the study and a recommendation.

Throughout the text, some words must match the format of the journal.

In general, the text has many errors that must be carefully corrected by the author.

Author Response: We have revised the text and corrected errors.

Use newer references

Author Response: We have provided newer and relevant references.

Reviewer #2

In this article, the concentration of some heavy metals in sesame seeds harvested from different farms in Ethiopia has been investigated. Although the topic of the article is a priority in terms of the importance of food health, but the article has flaws that are not acceptable in my opinion. These are listed below.

1- The introduction does not sufficiently specify the importance and innovation of this article.

Author Response: We have revised the introduction to better highlight the importance and novelty of the article.

2- Chemical fertilizers used in different regions are not mentioned in the materials and method section.

Author Response: We did not conduct any survey or study to investigate the nature and types of fertilizers used in the studied areas, as this was outside the scope of the research. Our focus was solely on examining the presence or absence of differences in trace metal concentrations in the sesame seeds, which is now clearly stated in the conclusion section.

3- It seems that checking and analyzing the concentration of heavy elements alone in sesame seeds is not enough. It is suggested that the concentration of heavy metals in the soil of the region was also checked.

Author Response: We did not investigate the relationship between the concentrations of metals in the sesame seeds and those in the soil, as this was outside the scope of the study. We believe that not all studies on plants require accompanying soil data, as it depends on the study's objectives. Our study focused on identifying variations in trace metal concentrations and assessing the associated health risks of sesame seeds from all commercial production areas in Ethiopia.

4- The results part seems to be more like presenting a scientific report rather than a scientific and research article.

Author Response: We have presented and discussed our findings in terms of the trace metal composition of sesame seeds, geographical variations across the study areas, comparison with existing literature, and the implications for health risks. We believe that the presentation meets the standards of a scientific paper.

5- What is the purpose of statistical correlation analysis? What is the conclusion of the presentation of correlation results?

Author Response: This is now clearly explained.

6- In Table 9, the correlation analysis results of significant levels are not specified.

Author Response: We have indicated the significant levels.

7- It is suggested to use more statistical analyzes such as principal component analysis in addition to correlation to determine the concentration of heavy metals.

Author Response: Principal component analysis has been included in the analysis of results.

8- The conclusion should be fundamentally rewritten. The results of this paper should be compared and evaluated with other similar researches in Ethiopia or other regions or other products.

Author Response: We have rewritten the conclusion. We had already compared and evaluated with other similar researches in Ethiopia or other countries.

9- What is the reason for the presence of heavy metals in sesame seeds? This is important to discuss.

Author Response: Our goal was to investigate the variation in trace metal content in sesame seeds from different cultivation regions of Ethiopia and assess the potential health risks associated with these variations. Determining the precise sources and their respective contributions would require a separate study.

Reviewer #3

The m

---

## [Decision Letter · Decision Letter 1]

5 Feb 2025

Trace metals and their human health risks in sesame seeds from the main cultivation areas of Ethiopia

PONE-D-24-46643R1

Dear Dr. Mehari,

We’re pleased to inform you that your manuscript has been judged scientifically suitable for publication and will be formally accepted for publication once it meets all outstanding technical requirements.

Kind regards,

Muhammad Sibt-e-Abbas

Academic Editor

PLOS ONE

Additional Editor Comments (optional):

Reviewers' comments:

Reviewer's Responses to Questions

**Comments to the Author**

1. If the authors have adequately addressed your comments raised in a previous round of review and you feel that this manuscript is now acceptable for publication, you may indicate that here to bypass the “Comments to the Author” section, enter your conflict of interest statement in the “Confidential to Editor” section, and submit your "Accept" recommendation.

Reviewer #1: All comments have been addressed

Reviewer #3: All comments have been addressed

2. Is the manuscript technically sound, and do the data support the conclusions?

Reviewer #1: Yes

Reviewer #3: Yes

3. Has the statistical analysis been performed appropriately and rigorously? 

Reviewer #1: Yes

Reviewer #3: Yes

4. Have the authors made all data underlying the findings in their manuscript fully available?

Reviewer #1: Yes

Reviewer #3: Yes

5. Is the manuscript presented in an intelligible fashion and written in standard English?

Reviewer #1: Yes

Reviewer #3: Yes

6. Review Comments to the Author

Reviewer #1: Accept and also there are no concerns about dual publication, research ethics, or publication ethics.

Reviewer #3: In my opinion, this new manuscript represents the authors' excellent work. The latest version of the manuscript has improved its quality.

Accept

7. PLOS authors have the option to publish the peer review history of their article (what does this mean? ). If published, this will include your full peer review and any attached files.

**Do you want your identity to be public for this peer review?** For information about this choice, including consent withdrawal, please see our Privacy Policy .

Reviewer #1: No

Reviewer #3: **Yes: ** Dr Mahmood Ahmed

---

## [Editor Report · Acceptance letter]

PONE-D-24-46643R1

PLOS ONE

Dear Dr. Mehari,

I'm pleased to inform you that your manuscript has been deemed suitable for publication in PLOS ONE. Congratulations! Your manuscript is now being handed over to our production team.

Kind regards,

on behalf of

Dr. Muhammad Sibt-e-Abbas

Academic Editor

PLOS ONE